# Transplanted Murine Tumours SPECT Imaging with ^99m^Tc Delivered with an Artificial Recombinant Protein

**DOI:** 10.3390/ijms251810197

**Published:** 2024-09-23

**Authors:** Natalia V. Pozdniakova, Alexey A. Lipengolts, Vsevolod A. Skribitsky, Kristina E. Shpakova, Yulia A. Finogenova, Anna V. Smirnova, Alexei B. Shevelev, Elena Y. Grigorieva

**Affiliations:** 1N.N. Blokhin National Medical Research Center of Oncology of the Ministry of Public Health of the Russian Federation (N.N. Blokhin NMRCO), Kashirskoe Shosse, 23, 115478 Moscow, Russia; lipengolts@mail.ru (A.A.L.); skvseva@yandex.ru (V.A.S.); shpakova.k.e@gmail.com (K.E.S.); b-f.finogenova@yandex.ru (Y.A.F.); smirn-ova@mail.ru (A.V.S.); grigelen@rambler.ru (E.Y.G.); 2N.I. Vavilov Institute of General Genetics RAS, Gubkina Street, 3, GSP-1, 119991 Moscow, Russia; shevel_a@hotmail.com; 3Institute of Engineering Physics for Biomedicine (PhysBio), National Research Nuclear University MEPhI, Kashirskoe Shosse, 31, 115409 Moscow, Russia

**Keywords:** SPECT, de novo protein design, technetium, targeted delivery, cancer, in vivo imaging

## Abstract

^99m^Tc is a well-known radionuclide that is widely used and readily available for SPECT/CT (Single-Photon Emission Computed Tomography) diagnosis. However, commercial isotope carriers are not specific enough to tumours, rapidly clear from the bloodstream, and are not safe. To overcome these limitations, we suggest immunologically compatible recombinant proteins containing a combination of metal binding sites as ^99m^Tc chelators and several different tumour-specific ligands for early detection of tumours. E1b protein containing metal-binding centres and tumour-specific ligands targeting integrin α_v_β_3_ and nucleolin, as well as a short Cys-rich sequence, was artificially constructed. It was produced in *E. coli*, purified by metal-chelate chromatography, and used to obtain a complex with ^99m^Tc. This was administered intravenously to healthy Balb/C mice at an activity dose of about 80 MBq per mouse, and the biodistribution was studied by SPECT/CT for 24 h. Free sodium ^99m^Tc-pertechnetate at the same dose was used as a reference. The selectivity of ^99m^Tc-E1b and the kinetics of isotope retention in tumours were then investigated in experiments in C57Bl/6 and Balb/C mice with subcutaneously transplanted lung carcinoma (LLC) or mammary adenocarcinoma (Ca755, EMT6, or 4T1). The radionuclide distribution ratio in tumour and adjacent normal tissue (T/N) steadily increased over 24 h, reaching 15.7 ± 4.2 for EMT6, 16.5 ± 3.8 for Ca755, 6.7 ± 4.2 for LLC, and 7.5 ± 3.1 for 4T1.

## 1. Introduction

### 1.1. ^99m^Tc Carriers for SPECT/CT Tumour Diagnosis

The history of tumour imaging using ^99m^Tc began with the labelling of tumour-specific monoclonal antibodies with this isotope, which required the addition of a linker and/or a ^99m^Tc chelator to the antibody. Since random chemical modification of antibodies can lead to a decrease in their specificity, various methods for site-directed modification of antibodies have been proposed. For example, in [1], the authors attached a ^99m^Tc-binding linker to humanised antibodies at well-defined protein glycosylation sites located outside the tumour ligand binding site.

In some studies, additional enzymatic modification of antibodies was required. Thus, in the study by Li et al. [2], an alternative approach was applied for site-directed labelling of humanised antibodies targeting the cancer marker HER2 (human epidermal growth factor receptor). They used synthetic G4EC-type peptides as ^99m^Tc chelator molecules, coupling them to the antibody by enzymatic transacylation with sortase A precisely at the site of the LPETG motif added to the antibody molecule. Enzymatic modification of antibodies along with the addition of a ^99m^Tc chelator was also used [3]. The authors used antibodies targeting tumour-associated macrophages. To improve pharmacokinetic characteristics, antibodies were hydrolysed with highly specific peptidases (papain or IdeS), followed by covalent attachment of a ^99m^Tc chelator to the resulting fragments. The examples provided [2,3] demonstrate that the use of specific antibodies allows for achieving the desired results, for example, providing effective visualisation of tumours in mice as early as 1 h after administration, but requires an additional step of chemical and/or enzymatic modification followed by purification.

In addition to massive protein molecules such as antibodies and their fragments, tumour-specific peptides, both naturally occurring and synthetic, are used for targeting. Synthetic peptides include, for example, somatostatin analogues, peptides targeting integrins, human epidermal growth factor receptor 2 (HER2), gastrin-releasing peptide receptor (GRPR), cholecystokinin receptors (CCK-1, CCK-2) and glucagon-like peptide-1 receptor (GLP-1) [4,5,6], and peptides targeting immune checkpoints PD-1/PD-L1 [7]. Many of them are currently in development and/or undergoing clinical trials. Some of them, in particular those containing the somatostatin analogue octreotide, have already been approved for Single-Photon Emission Computed Tomography (SPECT) Imaging of neuroendocrine tumours. Since these drugs are produced by peptide synthesis, the addition of an isotope-binding chelator occurs directly during the synthesis process. An example of a naturally occurring peptide is duramycin, a 19-amino acid antimicrobial peptide from the lantibiotic family produced by *Streptomyces cinnamoneus* and containing non-protein amino acids. Duramycin binds phosphatidylethanolamine exhibited on the outer side of the cell membrane, which is a hallmark of tumour cells [8]. Thus, in a study by Delvaeye et al. [9], SPECT imaging with ^99m^Tc-labelled duramycin was used to visualise inflammation caused by systemic administration of Tumour Necrosis Factor A (TNFα) to mice. ^99m^Tc labelling of duramycin also requires a preliminary step to obtain its derivative, hydrazinonicotinamide-(HYNIC) duramycin [10,11].

In addition to tumour-specific antibodies and peptides, the possibility of using serum albumin-based nanoparticles for targeted imaging of certain tumours is being studied. Thus, in [12], albumin nanoparticles labelled with ^99m^Tc with a diameter of less than 200 nm were used for SPECT imaging of murine multifocal hepatocarcinoma. The administered dose was 87.2 ± 8.3 MBq per animal. Standardized uptake value (SUV) of ^99m^Tc in the liver of different groups of animals varied from 6.2 to 11.2.

### 1.2. Known Types of Therapeutically Compatible ^99m^Tc Binding Groups

The presence of certain chemical groups in the structure of a molecule is necessary for labelling it with ^99m^Tc. For example, ^99m^Tc can bind to the SH-group of proteins and peptides, either naturally occurring or artificially introduced as part of a linker [13]. High-affinity chemical chelators have also been used to attach ^99m^Tc [14,15]. This also requires an additional step of chemical modification of the targeted molecule. A hexahistidine tag (His-tag), commonly incorporated into recombinant proteins for their purification by immobilised metal affinity chromatography (IMAC), can also be used to bind ^99m^Tc [16] in the ^99m^Tc-tricarbonyl ([^99m^Tc(CO)_3_]+) form. To label duramycin, a hydrazine group is introduced into its molecule, to which reduced ^99m^Tc, additionally stabilised by a chelator such as tricine, is then bound [10,11].

### 1.3. Artificial Metal-Binding Proteins as a Novel Approach to ^99m^Tc Delivery to Tumours

It can be assumed that there are still no generally available universal tools for the primary diagnosis of all tumour types by SPECT/CT. Duramycin derivatives contrast necrotic rather than actively proliferating tumour foci and often give false positive responses [17]. Monoclonal antibody derivatives and oncomarker-specific peptides recognise only certain tumour types, which is undesirable at the diagnostic stage when the tumour type and even its presence are unknown. Despite several advantages, tumour-specific peptides are not without disadvantages, including poor in vivo stability and a short half-life due to proteolytic degradation and rapid renal clearance. Complexes of ^99m^Tc with 6His motifs in proteins are unstable, leading to premature excretion of the isotope from the body before it reaches the tumour [16].

On the other hand, artificial immunologically compatible proteins combining several oncomarker ligands and high-affinity metal-binding sites derived from natural proteins (e.g., calmodulin repeats and/or clustered SH groups) have an advantage over all conventional carriers for primary SPECT/CT diagnosis of tumours due to their ability to specifically deliver the isotope to the tumour as part of a stable complex that is degraded in the target cell by proteolysis and therefore prevents the rapid evacuation of ^99m^Tc from the cancer cell into the bloodstream.

However, the design and production of such proteins is time-consuming, labour-intensive, and fraught with failure due to the complex structure and multivariate metabolism of the proteins in vivo. To our knowledge, our team is the only one that is currently developing such an approach to tumour theranostics. We hypothesise that all-protein ^99m^Tc carriers for SPECT/CT may be effective for longer post-injection exposures than is typical of current commercial preparations. In this case, the isotope delivered with the protein is better retained in tumour tissue than in adjacent non-tumour tissue, and therefore small tumour foci become more visually discernible. However, this approach requires higher doses of administered ^99m^Tc due to its short decay period, which is independent of the elimination rate.

In our previous study, we described the preparation of artificial W-family proteins containing a combination of multiple metal-binding centres and tumour-specific ligands—RGD (Arg-Gly-Asp, integrin α_v_β_3_-binding peptide) and F3-peptide (nucleolin-binding peptide) [18]. The protein elements were taken from the human proteome to ensure compatibility with the human body. The amino acid composition was carefully designed to prevent aggregation and non-specific binding in the bloodstream. We hypothesise that RGD/F3-containing proteins could potentially be used to develop novel ^99m^Tc-based radiopharmaceuticals for the diagnosis of primary tumours. In this study, a new recombinant protein, named E1b, was obtained by modifying one of the proteins (E2-13W4) in which Green Fluorescent Protein (GFP) was removed and replaced with a cysteine cluster to improve its binding to ^99m^Tc.

We aimed to evaluate the efficacy of ^99m^Tc-labelled E1b protein for tumour detection by SPECT/CT. We also considered the presence of a 7His-tag in the E1b protein, which may additionally contribute to ^99m^Tc binding [16]. To evaluate the ability of E1b to deliver ^99m^Tc to tumours in vivo, we used four mouse models of tumours (lung carcinoma LLC, mammary gland adenocarcinomas Ca755, EMT6, and 4T1). We calculated SUV for the tumour node and adjacent muscle and used the tumour-to-normal (T/N) ratio as a measure of diagnostic performance. Additionally, SUVs for all key organs (heart, lung, liver, kidney, bladder, intestine, stomach, muscle, brain, salivary, and thyroid glands) were evaluated to elucidate the biodistribution of the protein in the mouse body and to determine isotope excretion pathways. Free sodium pertechnetate was used as a reference in this study.

It should be noted that, unlike most of the quoted publications, our study was performed on mouse tumours and not on human tumours transplanted into mice with severe combined immunodeficiency (SCID). Although such an approach complicates extrapolating the data obtained to the diagnosis of real human tumours that have markers different from those in mouse tumours [19], it allows for the expansion of experimental cohorts, which is important in the early stages of testing candidate molecules. The use of murine tumour models allows for obtaining relevant data on pharmacokinetics and side effects of new molecules.

## 2. Results

### 2.1. Novel Recombinant Artificial Protein E1b

In this study, we introduce the ^99m^Tc carrier protein E1b for the first time. It is distinct from the previously described E2-13W4 protein in that it lacks the GFP fluorescent protein in the C-terminal position. Instead, it features a Cys-rich 12 amino acid flexible peptide (GlyGlyGlyCys)3 (further Cys-rich peptide) in that position. The E1b protein has a theoretically calculated MW = 25.7 kDa and contains functional elements specified in Table 1 (please refer also to Appendix B, Figure A1).

The overall structure of the E1b protein elements, the SDS-PAGE data (please refer also to Appendix A) demonstrating its purity, and the functional map of the pE1b plasmid (see also Appendix A) construct used for producing E1b in *E. coli* are depicted in Figure 1).

The E1b protein yield from the recombinant *E. coli* NiCo21 (DE3, pE1b) strain reached 18–20 mg/L culture, and it was present only in a soluble form. The apparent molecular weight (MW) of the E1b protein, as indicated by SDS-PAGE data (see Figure 1b), was approximately 40 kDa, which is significantly higher than the theoretical expectation of 25.7 kDa. We suspect that this discrepancy is due to the unusually low content of hydrophobic amino acids in comparison to most natural proteins. Following a simple process of immobilised metal ion affinity chromatography (IMAC) and dialysis, we obtained a solution of E1b protein with a concentration of 3.3 mg/mL (equivalent to 127.61 µM). According to Table 1, this solution contains 510.4 µM of potential coordination-type ^99m^Tc-binding sites and 382.3 µM of potential covalent-linkage sites. The electrophoretic purity of the preparation was determined to be 95%. In addition, the protein was found to contain approximately 390 µM of free sulfhydryl moieties as per the 5,5′-dithiobis-(2-nitrobenzoic acid) (DTNB) assay (refer to Section 4). Thanks to the described process, we were able to produce a protein preparation with high activity in a short period (Figure 2).

### 2.2. Biodistribution of ^99m^Tc-E1b Protein in Mice

The biodistribution of ^99m^Tc-E1b in healthy Balb/C mice was analysed using the SPECT/CT method in vivo (Table 2, Appendix C, Appendix A). The labelled protein was injected intravenously. To measure the compound’s presence in the blood, the average radioactivity in the heart chambers was calculated. Following intravenous administration, the compound was swiftly removed from the bloodstream: immediately after injection, the heart’s SUV reached 3.5 ± 0.3, then dropped rapidly to 0.6 ± 0.5 after 2 h, and decreased further to a background value of 0.1 ± 0.1 by 24 h. Immediately after the injection, there was a moderate uptake of ^99m^Tc-E1b in the lungs (SUV up to 1.8 ± 0.1) and in the liver (SUV up to 1.8 ± 0.2). Subsequently, within 24 h, the lung radioactivity returned to the background value while it remained slightly elevated in the liver (SUV 0.3 ± 0.1). By 6 h post-injection, a notable accumulation of radioactivity was observed in parts of the colon containing faeces (SUV up to 1.2 ± 0.1). Based on the data, there is evidence of at least partial hepatobiliary excretion of ^99m^Tc. High activity in the kidneys was detected from the first minute after administration (SUV 14 ± 1), with the radionuclide uptake primarily along the cortical layer of the kidneys. Activity in the kidneys increased over time and peaked at 2 h after injection (SUV 24 ± 6) (Table 2, Figure 3a), followed by a decrease to SUV 13 ± 1 by 24 h. At 3 h post-injection, high activity was also observed in the bladder (Table 2, Figure 3b), which suggests a predominantly renal route of ^99m^Tc excretion. There were no indications of ^99m^Tc-E1b accumulation in muscle tissue or in the brain.

Although only healthy mice were used for the quantitative biodistribution study, the SPECT data revealed a similar biodistribution pattern in both healthy and tumour-bearing mice.

### 2.3. Biodistribution of ^99m^Tc-Pertechnetate in Mice

The behaviour of free ^99m^Tc-pertechnetate and the synthesised ^99m^Tc-E1b complex in the bodies of Balb/C laboratory mice was evaluated using dynamic SPECT/CT (Table 2, Appendix A). ^99m^Tc-pertechnetate, a commonly used radiopharmaceutical drug for thyroid and salivary gland scintigraphy, as well as for the preparation of other radiopharmaceuticals, was chosen as a reference. After intravenous injection of ^99m^Tc-pertechnetate, the maximum activity in the heart chambers, reflecting the content of ^99m^Tc-pertechnetate in the blood, reached SUV 1.9 ± 0.3. This was followed by a gradual decrease, with the SUV decreasing to 0.7 ± 0.2 after 3 h and returning to the background level of 0.1 ± 0.1 after 24 h.

Significant accumulation of ^99m^Tc-pertechnetate in the neck area was observed (Table 2, Figure 3c), which increased over time. It is known from the literature that ^99m^Tc-pertechnetate accumulates in the thyroid gland [20] and, to a lesser extent, in the salivary glands [21]. However, it is difficult to precisely distinguish these organs based on SPECT/CT data due to their extremely small size in mice. The maximum accumulation in the neck area was observed 3 h post-injection (SUV 23 ± 3).

A significant amount of ^99m^Tc-pertechnetate accumulated in the pyloric part of the mouse’s stomach (Figure 3c), with the accumulation gradually increasing over time. The maximum accumulation was observed at 3 h post-injection (SUV 24 ± 3). The excretion of ^99m^Tc-pertechnetate by the gastric mucosa aligns with previously published data [22]. At 3 h post-injection, radioactivity accumulation was found in parts of the colon containing faeces (SUV up to 3.3 ± 0.5).

Immediately after injection, there was a moderate accumulation of ^99m^Tc-pertechnetate observed in the lungs (SUV up to 1.3 ± 0.2), liver (SUV up to 1.6 ± 0.3), and kidneys (SUV up to 1.4 ± 0.3) (Table 2, Figure 3c,d). After 24 h, lung activity returned to background levels, while activity in the liver and kidneys remained slightly elevated.

An intense accumulation of radioactivity was detected in the bladder, which increased over time and peaked at 3 h post-injection (SUV 10 ± 2), indicating the excretion of ^99m^Tc via the renal route. There were no signs of ^99m^Tc accumulation in muscle tissue or the brain.

### 2.4. Tumour Uptake of ^99m^Tc-E1b

The experiment used several transplantable tumour models that were fast-growing, easy to cultivate both in vivo and in vitro, well-characterised, and commonly used in experimental oncology. All of these selected tumour lines showed a high potential for spreading to other parts of the body and were likely to create an altered blood vessel network around the tumour. Additionally, previous studies have shown that the cell lines LL/2 (LLC1) [23], Ca755 [18], EMT6 [24], and 4T1 [25,26] have a tendency to accumulate specific ligands related to integrin α_V_β_3_ and nucleolin when tested in vivo after grafting. In all studied models, a rapid accumulation of ^99m^Tc-E1b in the tumour was observed in the first 2 h, followed by a gradual decrease within 24 h. However, the decrease in activity in the adjacent healthy tissues was faster, so the value of the tumour-to-normal tissue ratio (T/N) gradually increased during 24 h and reached the maximum at the end point of the observation period. The dynamics of T/N for ^99m^Tc-E1b in four syngeneic tumour models are shown in Table 3 and Figure 4 (see also Appendix A). At 24 h after drug administration, the highest T/N ratios were observed in EMT6 (15.7 ± 4.2) and Ca755 (16.5 ± 3.8) tumours. LLC (6.7 ± 4.2) and 4T1 (7.5 ± 3.1) tumours showed rather low accumulation and lower T/N ratios 24 h after administration.

Examples of SPECT/CT images of mice with all types of tumours 3 h after administration of the ^99m^Tc complex are presented in Figure 5.

## 3. Discussion

The E1b protein used in this study was derived from the previously described E2-13W4 protein [18] by replacing GFP with a C-terminal Cys-rich peptide. This protein has a completely artificial structure consisting of short motifs borrowed from human proteins (Table 1), which should ensure its immunological compatibility when administered to humans. Unlike most natural proteins, the E1b protein does not have a hydrophobic core. The introduced modifications did not affect the productivity of *E. coli* strains carrying the corresponding constructs: strains carrying both pE1b and pE2-13W4 showed protein yields at the level of 18–20 mg/L of bacterial culture. It is noteworthy that both proteins remained completely soluble at the time of synthesis; inclusion bodies containing the recombinant product were not formed under any culture conditions.

The use of recombinant E1b protein as a transport vehicle for delivery of ^99m^Tc to tumours for cancer diagnostics showed rather high efficiency. First of all, the high affinity of the protein to transition metals completely suppressed ^99m^Tc accumulation in the stomach, salivary glands, and thyroid gland, which is typical for free ^99m^Tc administered as sodium ^99m^Tc-pertechnetate (Table 2).

^99m^Tc bound to the E1b protein is excreted mainly through the kidneys (Table 2) but not through the gastric mucosa and salivary glands, which are the main routes of excretion of the isotope in the pertechnetate form. In most normal tissues, ^99m^Tc in any form is retained moderately (in the heart and lungs) or insignificantly (in the brain and skeletal muscles). The liver accumulates negligible amounts of the isotope in both cases, indicating that excretion via the bile is only a minor factor in the overall pharmacodynamics. In the heart and lungs, the E1b-bound isotope is retained somewhat faster and better than the isotope in the pertechnetate form. However, in both cases, the absolute SUV values are so small that they do not compromise the generally acknowledged toxicological safety of ^99m^Tc in SPECT/CT diagnostics [14]. These observations suggest that ^99m^Tc remains tightly bound to the protein during blood circulation. The revealed biodistribution of ^99m^Tc administered as a complex with E1b protein suggests that its use for tumour imaging will be highly effective for all tissues except kidney and, occasionally, liver tumours.

It should be noted that in our study, the T/N ratio for ^99m^Tc delivered in complex with E1b protein was lower in the case of 4T1 compared to other tumours. This is surprising since 4T1 breast carcinoma is the only one of all tumours we studied that contains both nucleolin and integrin α_V_β_3_ on the cell surface [25,26]. Theoretically, the E1b protein should have shown maximum targeting/retention ability on 4T1 cells because it can bind to both integrin α_V_β_3_ (RGD part of E1b protein) and nucleolin (F3 peptide part of E1b protein). However, it is important to note that the characteristics of grafted tumours can change over time, and we do not have experimental data on the amount of nucleolin and integrin α_V_β_3_ in both the 4T1 culture used in the experiment and other cell lines. In addition, there is evidence that specific obstacles in real tumours may interfere with the interaction of HER2-positive tumours with HER2-specific antibodies in animal models [2]. Similar mechanisms may also affect the use of ligand-specific isotope carriers in our and other cases. Given that most human tumours contain either integrins α_V_β_3_, nucleolin, or both of these markers on their surface, it is conceivable that a radiopharmaceutical based on E1b protein or its derivatives could be a rather versatile tool for primary cancer diagnosis by SPECT/CT. The protein labelling protocol proposed here demonstrates good reproducibility (see Appendix A), providing a high level of radioactivity of about 294 MBq per mg of E1b protein immediately after the purification step, as well as tight binding of ^99m^Tc to the protein.

Using the centrifuge-driven gel filtration for purification provides a quick removal of excess sodium ^99m^Tc pertechnetate, sodium borohydride, and other by-products in just 2 min, which is essential both to reduce the radiation dose received by personnel during manipulation and to maximise the specific activity of a radiochemical preparation with short-lived isotopes. The significant difference in molecular weight between the protein and the salts ensures reliable separation using this simple method, making it suitable for routine clinical practice. It is expected that the E1b protein will be applicable for SPECT diagnostics along with existing commercial ^99m^Tc carriers, offering advantages such as rapid clearance of most of the isotope in 6 h after administration and an ever-increasing ratio of tumour tissue activity to normal within 24 h. These pharmacokinetic properties have not been previously reported for any other ^99m^Tc carrier, which makes it a promising candidate for early detection of small tumour foci, provided that the CT scanners have sufficient sensitivity.

## 4. Materials and Methods

### 4.1. Chemicals and Disposables

NaBH_4_ (16940-66-2), LB broth (Luria low salt, L3397-250G), lysozyme from chicken egg, AEBSF (A8456-25MG), and other chemicals were obtained from Sigma-Aldrich (St. Louis, MO, USA). PD-10 gravity flow desalting column (17085101) was purchased from Cytiva (Muskegon, MI, USA). HisPur™ Ni-NTA Resin (88221), Zeba™ Spin Desalting Columns, 7K MWCO (89891) and the Pierce™ BCA Protein Assay Kit (23227) were purchased from Thermo Fisher Scientific (Waltham, MA, USA). Additionally, a sterile saline solution of 0.15 M NaCl (pH 7.4) was acquired from PanEco (Moscow, Russia). Sodium ^99m^Tc-pertechnetate was freshly eluted from the ^99m^Tc generator GT 4K manufactured by L.Y. Karpov NIPCI (Moscow, Russia).

### 4.2. Genetic Constructs

The pRSET-EmGFP plasmid (V35320, Thermo Fisher Scientific, Waltham, MA, USA) was used as a vector for the expression of the ^99m^Tc-carrier protein. High-fidelity restriction endonucleases (BamHI-HF, XhoI) and T4-DNA-ligase were purchased from New England Biolabs (Ipswich, MA, USA). The oligonucleotides were synthesised using the solid-phase method and purified by preparative polyacrylamide gel electrophoresis (PAGE) by Syntol LLC (Moscow, Russia). Q5^®^ High-Fidelity DNA Polymerase (New England Biolabs, Ipswich, MA, USA) was used for all preparative polymerase chain reactions (PCR). Ultrafree-DA Centrifugal Filter Units (42600, Merck, Rahway, NJ, USA) were used for DNA extraction from agarose gel. The ZymoPURE™ Plasmid Miniprep Kit (Zymo Research, Irvine, CA, USA) was used for plasmid DNA purification. The authenticity of the plasmids was confirmed by Sanger sequencing performed by Eurogen CJSC (Moscow, Russia). *E. coli* strain NiCo21(DE3) (C2529H, New England Biolabs, Ipswich, MA, USA) was used for cloning and expression experiments according to the manufacturer’s protocol.

### 4.3. Software

SnapGene Viewer (GSL Biotech LLC, Chicago, IL, USA, available at https://www.snapgene.com/) accessed on 18 July 2023 was used to design and manage plasmid maps. Oligo Analyzer 1.0.3, a free software program developed by T. Kuulasmaa, was used to design oligonucleotides. GelAnalyzer 19.1, a free software program developed by I. Lázár, was used for digital gel densitometry. Protein properties were predicted by using the Protein Calculator v3.4 free online tool (https://protcalc.sourceforge.net/, accessed on 18 July 2023). Image reconstruction (SPECT and CT) was performed using the built-in MiLabs Rec 12.00 software. The processing of SPECT/CT images was carried out using PMOD 4.205 software (PMOD Technologies LLC, Zurich, Switzerland).

### 4.4. Murine Tumour Cell Cultures

Cell lines Lewis lung adenocarcinoma (American Types of Culture Collection (ATCC) CRL-1642), mammary adenocarcinoma Ca755 (or Bagg-Jackson adenocarcinoma), metastatic mammary adenocarcinoma EMT6 (ATCC CRL-2755™), mammary adenocarcinoma 4T1 (ATCC CRL-2539™) were obtained from the Blokhin National Medical Research Centre of Oncology of the Ministry of Health of the Russian Federation (Blokhin NMRCO) cell collection. Details about the model cell lines utilised in the study can be found in Table 4.

### 4.5. Production and Purification of E1b Protein

#### 4.5.1. Genetic Construct pE1b

The plasmid pE2-13W4, previously described [18], was used as the source. Initially, a 4067 bp long PCR product was obtained using the pE2-13W4 plasmid as a template and a pair of primers: GFPmin-for cgaccacatgaagcagcacgac and GFPminBam-rev ccatggtggcgaaggatccgctaccagg. Thus, the BamHI restriction site was introduced at the end of the coding region. After purification, the PCR product was cleaved with BamHI HF and XhoI restriction enzymes following the manufacturer’s protocol. A 3558 bp long BamHI/XhoI PCR product was purified from an agarose gel after preparative electrophoresis.

An insert encoding a 12 amino acid Cys-containing flexible peptide (GGGCGGGCGGGC) with subsequent stop codon was assembled from a pair of complementary oligonucleotides: C3-for gatccggtggcggttgcggtggcggttgtggtgttgttaac and C3-rev tcgagttaacaaccgccaccacaaccgccaccgcaaccgccaccg (Syntol, Moscow, Russia), generating sticky ends for restriction sites BamHI and XhoI upon duplex formation. It was then ligated with the pre-prepared vector described in the previous paragraph. The *E. coli* strain NiCo21(DE3), recommended for efficient expression, was immediately transformed with a DNA mix after ligation. The sequence of the E1b protein gene in the expression construct was confirmed by restriction mapping, PCR with specific primers, and Sanger sequencing (Eurogen, Moscow, Russia).

#### 4.5.2. E1b Protein Production and Purification

The recombinant *E. coli* strain derived from NiCo21 (DE3, pE1b) was cultured in a complete LB medium (composed of 10 g/L peptone, 5 g/L yeast extract, and 10 g/L NaCl), supplemented with 100 mg/L ampicillin, for 24 h at 37 °C. The harvested cells were lysed and used to purify the recombinant protein, following the previously described method [18]. The purification process involved metal chelate chromatography (IMAC) using HisPur™ Ni NTA resin and the removal of metal ions from metal-binding sites through extensive dialysis after Ethylenediaminetetraacetic Acid (EDTA) treatment. The purity of the E1b protein preparation was confirmed using denaturing disc electrophoresis in a polyacrylamide gel with sodium dodecyl sulphate (SDS-PAGE). The protein concentration was determined using a modified Lowry method with a bicinchoninic acid (BCA) assay kit from Sigma, following the manufacturer’s recommendations. The presence of free sulfhydryl groups in the protein preparation was evaluated by reacting with DTNB as described previously [34].

### 4.6. Labelling E1b Protein with ^99m^Tc and Purification of the Complex

The experiment was conducted in a specially designed enclosure with a protective screen to adhere to safety regulations for handling radioactive materials. During the study, the ^99m^Tc labelling procedure was repeated several times, and the resulting preparation could vary slightly in radioactivity (refer to Appendix A). We standardised the administered dose (~80 MBq for each mouse) by slightly varying the injection volume of the preparation. The typical labelling protocol is described below. Initially, 280 µL of a ^99m^Tc solution (~995.4 MBq), 0.018 g (0.476 mmol) of NaBH_4_, and 700 µL (2.3 mg) of protein E1b were mixed to achieve a final volume of 1 mL. The mixture was then left to incubate for 30 min at room temperature (18–20 °C) before being desalted using Zeba™ Spin Desalting Columns with a 7K MWCO, as per the manufacturer’s instructions, using sterile saline at pH 7.2. The radioactivity of the eluate was immediately measured, and approximately 80 MBq (about 120 µL) of the eluate was injected intravenously into a mouse. A control group of mice was administered an equivalent radioactivity of free sodium ^99m^Tc-pertechnetate.

### 4.7. Tumour Animal Model

All the animal studies were conducted in compliance with local ethical regulations and were approved by the institutional ethics committee (Protocol No. 2, dated 10 June 2020). The animals were maintained in accordance with the rules of the European Convention for the Protection of Vertebrates Used for Research and Other Scientific Purposes [35]. For all in vivo studies, female laboratory mice of the C57Bl/6 and Balb/c lines aged 6–8 weeks and weighing 20–22 g (bred by Blokhin NMRCO) were used. To study the biodistribution of recombinant protein and free ^99m^Tc-pertechnetate as a reference in healthy mice, 2 groups of Balb/c mice (*n* = 6) were used. Additionally, 4 groups of tumour-bearing mice (*n* = 3 in each group) were formed to assess the accumulation of ^99m^Tc-E1b protein in solid tumours: (1) C57Bl/6 mice with lung adenocarcinoma LLC, (2) C57Bl/6 mice with mammary adenocarcinoma Ca755, (3) Balb/C mice with mammary adenocarcinoma 4T1, (4) Balb/C mice with breast adenocarcinoma EMT6. The inoculation of all tumour strains was carried out as per [36]. Ca755 and EMT6 tumours were grafted into the right hind leg, and LLC and 4T1 into the right side. In total, 24 mice were used in the experiment.

During scanning, the animals were anaesthetised with a 2% isoflurane air mixture. The animals’ condition was monitored by assessing respiratory rate using built-in equipment and BioVet Rev 04 software. Post the SPECT/CT studies, the mice were kept in a vivarium with daily body weight measurements and a general veterinary examination, along with monitoring water and feed consumption. Changes in body weight were consistent with tumour growth, and no changes in water and feed consumption and behaviour were noted.

### 4.8. SPECT/CT In Vivo Imaging

The substance’s radioactivity was measured using the Isomed 2010 dose calibrator (MED Nuclear Medizintechnik Dresden Gmb, Dresden, Germany) before it was administered. A small laboratory animal Vector 6 (MiLabs, Houten, The Netherlands) PET/SPECT/CT scanner was used to perform the SPECT/CT study with a HE-UHR-RM collimator. Both ^99m^Tc-E1b protein and ^99m^Tc-pertechnetate compounds were injected intravenously into mice. SPECT scanning began 5 min after the ^99m^Tc administration and continued for 1 h in 12 frames of 5 min each. Time points at 2 h, 3 h, 6 h, and 24 h after administration were scanned in frames of 10 min, 10 min, 15 min, and 30 min, respectively. For ^99m^Tc-pertechnetate, the first-hour protocol was the same, and then the data at only 3 h and 24 h were acquired. The image reconstruction (SPECT and CT) was done using the built-in MILabs Rec 12.00 software. For SPECT reconstruction, the following parameters were chosen: energy window 140 ± 10% keV, SROSEM iterative algorithm, voxel size of 0.8 mm, and radioactive decay correction function. CT images were reconstructed using the Radon transform with a voxel size of 0.2 mm. Subsequently, the software MILabs Rec 12.00 registered pairs of SPECT and CT images and produced merged SPECT/CT images.

### 4.9. Processing of SPECT/CT Data

The SPECT/CT images were processed using PMOD 4.025 software (PMOD Technologies Ltd., Zürich, Switzerland). We outlined volumes of interest using the CT image of the mouse and calculated the average activity within these outlined volumes (measured in MBq/cc). For the biodistribution study, we focused on the following organs: heart, lungs, liver, kidneys, bladder, bowel, muscle, and brain. Additionally, for the biodistribution of ^99m^Tc-pertechnetate, we also examined the stomach and thyroid gland. In mice with tumours, we outlined the tumours, heart, and muscle for a quantitative study. Total body radioactivity was determined for all animals. We calculated the standardised uptake value (SUV) for all the studied organs and tissues. To provide a statistical estimate of the SUV value for each organ, we evaluated the mean value for all the animals in each group, with a confidence interval of 95% as ±SD (Standard Deviation).

## 5. Conclusions

The use of artificially created E1b protein as a carrier of ^99m^Tc for imaging of transplanted murine tumours LLC (lung carcinoma), 4T1, Ca755, and EMT6 (mammary carcinoma) allowed us to obtain a marked difference in isotope concentration in tumour and adjacent healthy tissues 24 h after injection, despite the fact that the total isotope accumulation in tumours was rather low, especially in the case of 4T1 and LLC tumours. This observation indicates the validity of our hypothesis of longer retention in tumours of ^99m^Tc, as well as other radionuclides delivered in complex with protein (rather than with chemical ligands resistant to degradation in the tumour cell). In our opinion, this is the key advantage of protein carriers of ^99m^Tc, making them promising for primary diagnosis of small tumour foci by SPECT/CT, as well as for the development of theranostic radiopharmaceuticals.

## Figures and Tables

**Figure 1 ijms-25-10197-f001:**
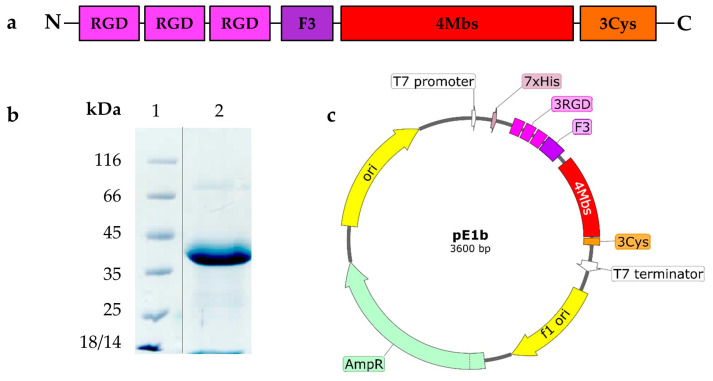
E1b artificial protein, its production, and purification. (**a**) Block-scheme of functional elements composing E1b protein; (**b**) electropherogram of the purified E1b protein used as ^99m^Tc carrier in biological trials (electrophoresis was carried out in 12.5% Polyacrylamide gel supplemented with SDS and stained with Coomassie R-250; lanes: 1—molecular mass standard (Fermentas, Lithuania), 2—purified E1b (vertical black line indicates spliced lines of the same gel)); (**c**) plasmid map pE1b, generated with SnapGene Viewer; Element legend—3RGD—three tandem RGD-containing peptides, F3—F3 peptide, 4MBS—four metal-binding sites alternating with ELP-repeats, 3Cys—C-terminal Cys-rich peptide.

**Figure 2 ijms-25-10197-f002:**
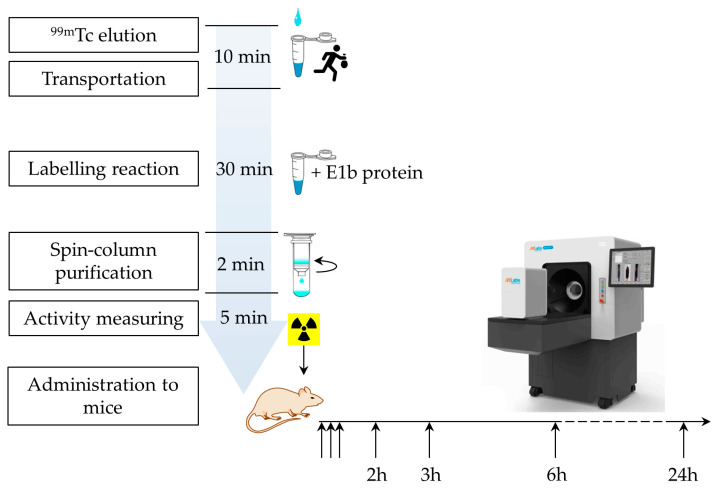
Timing of the protein labelling procedure and the in vivo experiment.

**Figure 3 ijms-25-10197-f003:**
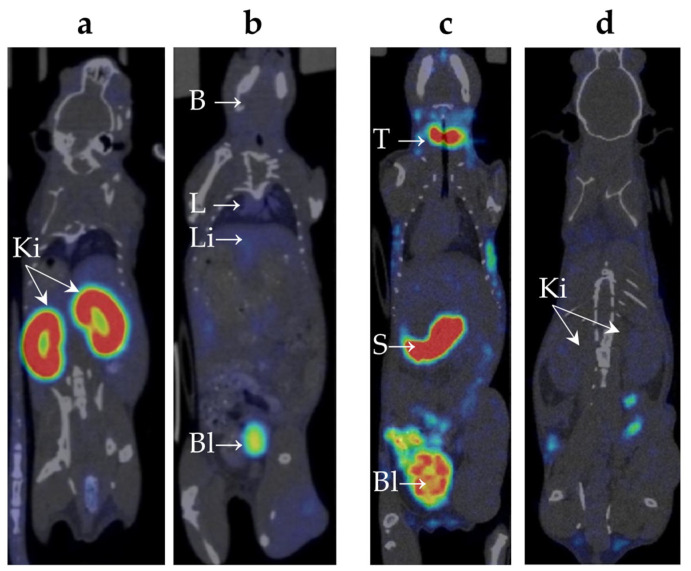
Single-Photon Emission Computed Tomography (SPECT/CT) scan of the healthy mice 3 h after intravenous administration of the radiopharmaceutical compound (^99m^Tc-E1b (**a**,**b**) and ^99m^Tc-sodium pertechnetate (**c**,**d**)). 2D scans for each organ group were chosen to maximise their visualisation. The images taken after administration of ^99m^Tc-E1b show strong accumulation in the kidneys, mainly in the renal cortex (**a**), low accumulation in the lungs and liver, no accumulation in the stomach, and strong accumulation in the bladder (**b**). Images taken after ^99m^Tc-pertechnetate administration show strong accumulation in the thyroid gland, stomach, and bladder, low accumulation in the liver (**c**), and low accumulation in the kidneys (**d**). The labels used are B for brain, L for lung, Li for liver, Bl for urinary bladder, Ki for kidney, T for thyroid gland, and S for stomach.

**Figure 4 ijms-25-10197-f004:**
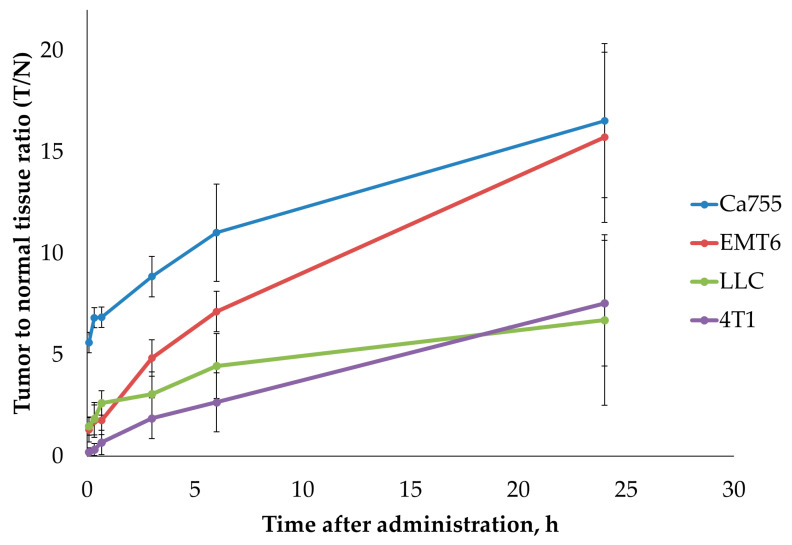
Dynamics of the ratio of the radioactivity in tumours to the radioactivity in muscles as a function of time after administration of ^99m^Tc-E1b in four transferrable tumour models. The plot specifies the 5, 20, 40 min, 3, 6, and 24 h time points. Each value displayed on the graph is an average SUV value obtained from three animals. The vertical bars represent the standard deviation values.

**Figure 5 ijms-25-10197-f005:**
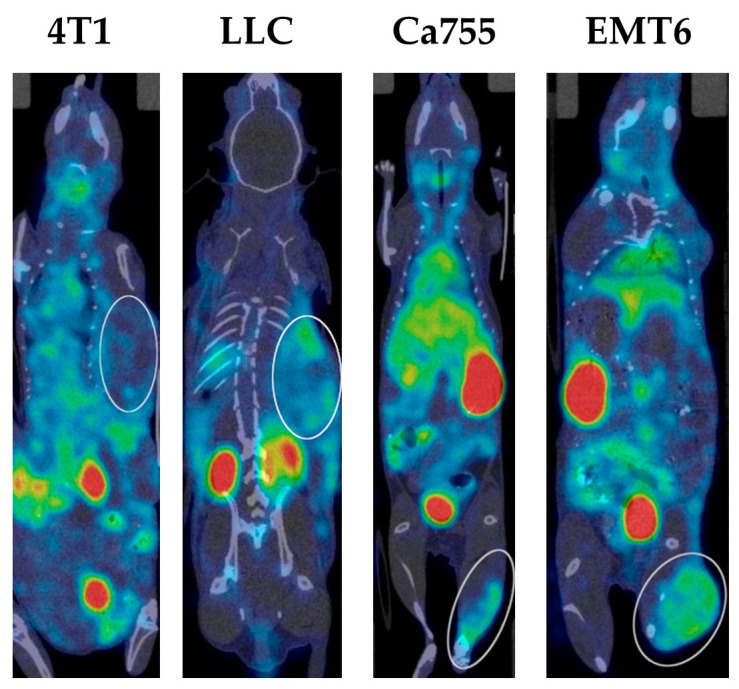
An example of SPECT/CT scan of mice with transplanted tumours at 3 h after ^99m^Tc-E1b intravenous administration. The location of 4T1, LLC, Ca755, and EMT6 tumours is indicated by the white ellipse.

**Table 1 ijms-25-10197-t001:** Stoichiometry of functional elements in E1b ^99m^Tc carrier protein.

Functional Elements	Number per Molecule	Function
His-taq	1	Protein purification by IMAC
RGD-motif	3	Binding integrin α_V_β_3_
F3-peptide	1	Nucleolin ligand binding
Metal binding site (Mbs)	4	Radionuclide binding in coordinated complex
Free cysteine	3	Radionuclide covalent coupling
ELP repeats (5 amino acids)	13	Immune-compatible molecular framework

**Table 2 ijms-25-10197-t002:** Dynamics of ^99m^Tc biodistribution administrated to healthy mice in the complex with E1b protein and the form of sodium ^99m^Tc-pertechnetate.

Organ	Time Point	^99m^Tc-E1b Protein (SUV)	^99m^Tc-Pertechnetate (SUV)
Heart	5 min	3.5 ± 0.5	2.1 ± 0.3
	20 min	2.1 ± 0.3	1.2 ± 0.2
	40 min	1.4 ± 0.4	0.8 ± 0.2
	2 h	0.6 ± 0.3	
	3 h	0.4 ± 0.2	0.7 ± 0.1
	6 h	0.1 ± 0.1	
	24 h	0.1 ± 0.1	0.05 ± 0.01
Lungs	5 min	1.8 ± 0.2	1.3 ± 0.3
	20 min	1.2 ± 0.4	0.9 ± 0.2
	40 min	0.8 ± 0.3	0.6 ± 0.2
	2 h	0.3 ± 0.2	
	3 h	0.2 ± 0.2	0.4 ± 0.1
	6 h	0.07 ± 0.02	
	24 h	0.04 ± 0.02	0.04 ± 0.02
Liver	5 min	1.8 ± 0.2	1.6 ± 0.3
	20 min	1.3 ± 0.2	1.5 ± 0.4
	40 min	1.0 ± 0.1	1.4 ± 0.2
	2 h	0.7 ± 0.1	
	3 h	0.6 ± 0.1	0.8 ± 0.2
	6 h	0.5 ± 0.1	
	24 h	0.3 ± 0.1	0.1 ± 0.1
Kidneys	5 min	14 ± 1	1.4 ± 0.3
	20 min	18 ± 2	0.9 ± 0.3
	40 min	18 ± 1	0.7 ± 0.2
	2 h	24 ± 6	
	3 h	22 ± 5	0.6 ± 0.1
	6 h	22 ± 7	
	24 h	13 ± 1	0.1 ± 0.1
Stomach	5 min	0.5 ± 0.1	5 ± 1
	20 min	0.3 ± 0.1	7 ± 1
	40 min	0.2 ± 0.07	12 ± 2
	2 h	0.1 ± 0.05	
	3 h	0.1 ± 0.05	24 ± 2
	6 h	0.08 ± 0.01	
	24 h	0.05 ± 0.01	1.4 ± 0.3
Neck (thyroid and salivary gland area)	5 min	0.2 ± 0.1	5 ± 2
	20 min	0.2 ± 0.1	9 ± 2
	40 min	0.1 ± 0.1	12 ± 1
	2 h	0.1 ± 0.1	
	3 h	0.06 ± 0.04	23 ± 3
	6 h	0.02 ± 0.01	
	24 h	0.02 ± 0.01	0.9 ± 0.2
Muscle	5 min	0.3 ± 0.1	0.3 ± 0.1
	20 min	0.2 ± 0.1	0.2 ± 0.2
	40 min	0.17 ± 0.07	0.2 ± 0.1
	2 h	0.07 ± 0.05	
	3 h	0.05 ± 0.05	0.1 ± 0.1
	6 h	0.03 ± 0.01	
	24 h	0.02 ± 0.01	0.01 ± 0.01

**Table 3 ijms-25-10197-t003:** The tumour-to-normal tissue ratio (T/N) dynamics of ^99m^Tc-E1b in four syngeneic tumour models.

T, min	Ca755	EMT6	LLC	4T1
2	5.6 ± 0.5	1.3 ± 0.6	1.5 ± 0.5	0.2 ± 0.2
20	6.8 ± 0.5	1.7 ± 0.8	1.8 ± 0.8	0.3 ± 0.3
40	6.8 ± 0.5	1.8 ± 0.7	2.6 ± 0.6	0.7 ± 0.6
180	8.9 ± 1.1	4.8 ± 0.9	3.1 ± 1.1	1.9 ± 1.0
360	11.0 ± 2.4	7.1 ± 1.0	4.4 ± 1.6	2.7 ± 1.5
1440	16.5 ± 3.8	15.7 ± 4.2	6.7 ± 4.2	7.5 ± 3.1

**Table 4 ijms-25-10197-t004:** Data about model cell lines used in the study.

Name	ATCC No	Properties	Protocol of Maintenance	Immune Compatibility with Mouse-Inbred Lines	Peculiarities
LL/2 (LLC1)	CRL-1642™	Transplantable lung Lewis adenocarcinoma occurs as three clones Lab, LLCC3 and LLC1. In our study, the clone LLC1 CRL 1642 was used	[27,28]	C57Bl/6	A traditional model of metastatic growth for the study of resistance to chemotherapeutic agents
Ca755 or Bagg-Jackson Adenocarcinoma	-	Transplantable mammary gland adenocarcinoma of C57/Bl mice	[29]	C57Bl/6	An acknowledged model of triple-negative human breast cancer with a high metastatic potential
EMT6	CRL-2755™	Metastatic mammary adenocarcinoma	[30,31,32]	Balb/c	https://www.atcc.org/products/crl-2755/ (accessed on 18 July 2023) The rapid growth of the tumour node begins from 12–15 days after grafting
4T1	CRL-2539™	Mammary adenocarcinoma	[33]	Balb/c	https://www.atcc.org/products/crl-2539/ (accessed on 18 July 2023) popular model of the human TNM stage IV

## Data Availability

The data that support the findings of this study are included in this published article and its Appendix A.

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
