# Peer review of "Transplanted Murine Tumours SPECT Imaging with 99mTc Delivered with an Artificial Recombinant Protein"

_ijms, 2024, doi:10.3390/ijms251810197_

Round 1

Reviewer 1 Report

Comments and Suggestions for Authors

The authors radiolabeled an artificial recombinant protein E1b with 99mTc. Then 99mTc-E1b was evaluated by SPET/CT and biodistribution experiments. This article is logical and remains of high interest to the field. However, 99mTc-E1b cannot serve as a promising radioligand in tumor imaging.

Main Comments:

1. What is the background of E1b? The authors should give more details about this protein, including its targets and functions.

2. The binding affinity between E1b and its target should be evaluated. Surface Plasmon Resonance (SPR) or Isothermal Titration Calorimetry (ITC) experiments are recommended to measure the Kd values.

3. In Figure 4, there is no obvious tumor in leg or breast, please provide more information about the tumor, such as tumor sizes and tumor pictures.

Minor comments:

1. Line 64 “Enzymatic modification of antibodies was also described in [6]” What is [6]?

2. Line 77 “Duramycin is a 19 a.a. tetracyclic antimicrobial” a.a should be “amino acid”

3. “In vivo” and “in vitro” should be italic.

4. The manuscript should be checked carefully in terms of errors (very numerous throughout the text). The manuscript requires also editorial improvement.

Comments on the Quality of English Language

The manuscript should be checked carefully in terms of errors (very numerous throughout the text). The manuscript requires also editorial improvement

"

Reviewer 2 Report

Comments and Suggestions for Authors

The paper is focused on SPECT imaging of transplanted murine tumors using 99mTc delivered with an artificial recombinant protein.  The results demonstrate that the use of the E1b protein significantly increased the attraction of 99mTc to experimentally transplanted mouse tumors. The manuscript has been thoroughly prepared. The results are understandable, and they support the conclusions. However, I am of the opinion that the discussion can be improved by thoroughly dealing with the mentioned concerns and incorporating each point in the manuscripts prior to publication.

1 - Has the thermodynamic stability of 99mTc-E1b protein been assessed? Is 99mTc- chemically bound to the-E1b protein, to ensure optimal stability and activity in the biological environment at an optimal level? An illustration in 3D of the  99mTc-E1b protein  where we visualize the different motifs as well as the part which chemically binds 99mTc-, is recommended. Are the 99mTc-E1b protein's luminiscence properties present and being utilized?

2- To target different tumors, it is necessary to have clarity on the manifestation of biological resistance, which could include autophagy, apoptosis dysregulation, epigenetic alteration, gene mutation or amplification, and so on.

3 - Is the size of the E1b protein optimal compared to small molecules that can cross several biological barriers to reach tumors?

4- Are the doses administered optimal or can they be optimized to obtain better results? What protocols have you established to support the doses injected?

5 - Through kinetic analysis of the dynamic of 99mTc-E1b protein, specific interactions between the radiotracer and tissue can be quantified. A kinetic model should be capable of accurately estimating kinetic parameters that have biological relevance. The absence of these aspects hinders the full assessment of tumors and the identification of possible mutations. It is recommended that the authors evaluate nad discuss this point.

Reviewer 3 Report

Comments and Suggestions for Authors

This manuscript presents a well-executed study where the authors designed a protein, E1b, containing a metal-binding center with ELP repeats, tumor-specific ligands targeting integrin αvβ3 and nucleolin, and a short Cys-rich sequence. The protein was produced in E. coli. It was then purified using metal-chelate chromatography. The selectivity and retention kinetics of 99mTc-E1b were evaluated in C57Bl/6 and Balb/C mice with subcutaneously grafted lung carcinoma (LLC) or mammary gland adenocarcinoma (Ca755, EMT6, or 4T1). For these tests, approximately 80 MBq of 99mTc complexed with freshly purified E1b was intravenously injected into the mice. The manuscript is suitable for publication. However, the authors are requested to provide a better rationale for the novelty aspect. The authors may consider slightly shortening the introduction part too. Figure 5 was studied only for 24 hours. Was there a particular reason for this choice of duration? 

Comments on the Quality of English Language

No problem.

Reviewer 4 Report

Comments and Suggestions for Authors

Please revise the manuscript to maintain the abbreviation of liter (sometimes it is l and sometimes L). The absent Figures S2-S4 are needed to evaluate the manuscript fully, it is not possible now to review it correctly.  

Line 64: It should state “Enzymatic modification of antibodies was also described by Shi et al.”

A huge part of the introduction discusses various aspects of Tc, especially paragraph 1.2. I recommend shortening the introduction and moving it to discuss the results since the discussion is rather short.

Line 217: The figure can be improved. As I understood this is 99mTc distribution in mouse after 3 hours. Why do separate images show separate accumulations? What is the factor differentiating a, b, c and d?

Line 272: The description of Tc-E1b uptake of neoplasms is very poor. EMT6 and Ca755 seem to be visualized well, but LLC and 4T1 are not so good. A good description of the results is lacking. Also, I believe that the control group should be presented alongside to indicate the Tc uptake by healthy organisms.

Can you explain why in the lungs it is more concentrated and in other in kidneys? It should be discussed, as the results are not the same for different tumors

Lines 301-308: the table and the description should be moved to the results section

Line 392: What was the transplantation procedure of tumors?

Line 478: 10 minutes is doubled

The manuscript lacks a conclusions section to summarize the study.

In the supplementary materials only Figure S1 is provided, referenced materials of S2-S4 are not possible to verify, which should be included in the revised manuscript. 

Comments on the Quality of English Language

The English language is very good, and readability and scientific nomenclature are correct. 

Round 2

Reviewer 1 Report

Comments and Suggestions for Authors

The revised manuscript resolved my concerns, it can be accepted in current form.

Reviewer 4 Report

Comments and Suggestions for Authors

Dear authors,

Thank you for revising the manuscript, my comments were considered and questions were addressed competently. I still feel that the introduction can be shortened, however, the manuscript quality level meets the criteria to be considered for publication.